# Liver Transplantation for HCC in HIV-Infected Patients: Long-Term Single-Center Experience

**DOI:** 10.3390/cancers13184727

**Published:** 2021-09-21

**Authors:** Gian Piero Guerrini, Massimiliano Berretta, Giovanni Guaraldi, Paolo Magistri, Giuseppe Esposito, Roberto Ballarin, Valentina Serra, Stefano Di Sandro, Fabrizio Di Benedetto

**Affiliations:** 1Hepato-Pancreato-Biliary Surgery and Liver Transplantation Unit, University of Modena and Reggio Emilia, Azienda Ospedaliero-Universitaria di Modena, 41125 Modena, Italy; guerrini.gianpiero@aou.mo.it (G.P.G.); paolo.magistri@unimore.it (P.M.); giuseppe.esposito9191@gmail.com (G.E.); ballarin.roberto@aou.mo.it (R.B.); serra.valentina@aou.mo.it (V.S.); stefano.disandro@unimore.it (S.D.S.); 2Infectious Diseases Unit, Department of Clinical and Experimental Medicine, University of Messina, 98122 Messina, Italy; massimiliano.berretta@unime.it; 3Infectious Diseases Unit, Department of Surgical, Medical, Dental and Morphological Sciences, University of Modena and Reggio Emilia, Azienda Ospedaliero-Universitaria di Modena, 41125 Modena, Italy; giovanni.guaraldi@unimore.it

**Keywords:** AFP = alpha-fetoprotein, CI = confidence interval, HCC = hepatocellular carcinoma, HR = hazard ratio, LT = liver transplantation, MELD = model for end-stage liver disease

## Abstract

**Simple Summary:**

HCC are now emerging as leading causes of morbidity in HIV-infected patients since HAART has made it possible to achieve long life expectation by reducing AIDS-related complications. The results of liver transplantation for HCC in HIV-infected patients are still contradictory. In this paper we demonstrated in a cohort of 32 HIV-infected patients undergoing LT for HCC that the post-LT outcomes are comparable to the controlled cohort of uninfected recipients. Long survival rates were observed for HIV-infected LT patients with HCC.

**Abstract:**

Background: HIV-infected patients now have long life expectation since the introduction of the highly active antiretroviral therapy (HAART). Liver diseases, especially cirrhosis and hepatocellular carcinoma (HCC), currently represent a leading cause of death in this setting of patients. Aim: To address the results of liver transplantation (LT) for HCC in HIV-infected patients. Methods: All patients with and without HIV infection who underwent LT for HCC (n = 420) between 2001 and 2021 in our center were analyzed with the intent of comparing graft and patient survival. Cox regression analysis was used to determine prognostic survival factors and logistic regression to determine the predictor factors of post-LT recurrence. Results: Among 1010 LT, 32 were HIV-infected recipients. With an average follow-up of 62 ± 51 months, 5-year overall survival in LT recipients with and without HIV-infection was 71.6% and 69.9%, respectively (*p* = ns), whereas 5-year graft survival in HIV-infected and HIV-non infected was 68.3% and 68.2%, respectively (*p* = ns). The independent predictive factor of survival in the study group was: HCV infection (HR 1.83, *p* = 0.024). There were no significant differences in the pathological characteristics of HCC between the two groups. The logistic regression analysis of the study population demonstrated that microvascular invasion (HR 5.18, *p*< 0.001), HCC diameter (HR 1.16, *p* = 0.028), and number of HCC nodules (HR 1.26, *p* = 0.003) were predictors of recurrence post-LT. Conclusion: Our study shows that HIV patients undergoing LT for HCC have comparable results in terms of post-LT survival. Excellent results can be achieved for HIV-infected patients with HCC, as long as a strategy of close surveillance and precise treatment of the tumor is adopted while on the waiting list.

## 1. Introduction

The introduction of highly active antiretroviral therapy (HAART) has revolutionized the long-term survival expectations of HIV patient [1]. For this reason, long-term morbidity currently appears to be linked to the onset of chronic diseases, while the risk of mortality from opportunistic diseases and AIDS-related events is now dramatically reduced [2]. With longer survival, liver diseases including chronic viral hepatitis and hepatocellular carcinoma (HCC) are frequently observed illnesses in this population [3,4]. In addition, due to shared transmission routes with HIV infection, HBV and HCV increase the risk of developing chronic liver diseases. Moreover, HIV coinfection has been shown to increase the progression of liver fibrosis. In fact, end stage liver disease (ESLD) and HCC are currently the most frequent causes of death in the HIV-infected population, with an annual risk of liver-related death mortality of 50% [5].

Liver transplantation has proved to be a well-established cure in the treatment of chronic liver diseases as well as in liver tumors [6]. Historically, HIV patients have been denied access to liver transplantation for a long time due to worse outcomes than the uninfected population. However, the better understanding of the immunological and infectious status and the use of precise selection criteria of these patients have made it possible to achieve very encouraging results [7,8,9].

HCC is a tumor whose incidence is growing worldwide [10]; in this population, HCC is an emerging problem as HIV patients with cirrhosis have been shown to have a higher risk of tumor transformation than the uninfected population due to the specific oncogenesis pathway. Liver transplantation for HCC in the HIV-infected population is still debated since it is believed that this tumor tends to manifest with more invasive and aggressive forms that can increase the risk of recurrence after LT [11,12].

The aim of this study is to assess the outcome and long-term survival of our cohort of HIV-infected patients undergoing liver transplantation for HCC. We therefore compared HIV transplant patients with the historical control group of HCC patients transplanted in the same period with the aim of identifying prognostic factors of survival and recurrence after LT.

## 2. Materials and Methods

### 2.1. Study Population

A prospectively collected database of all the 1010 liver transplantations performed in 946 patients at Modena University Hospital from December 2001 through June 2021 was retrospectively reviewed. Over the 20-year period, 420 LT for HCC were performed in 400 patients; 34 of these liver transplants (in 32 patients) had an HIV-infection.

HIV-infected liver transplant candidates were evaluated by a multidisciplinary team (consisting of surgeons, hepatologists, infectious disease physicians, and radiologists), with the aim of verifying that patients met immunological and infectious criteria for inclusion in the transplant program.

The inclusion criteria for HIV patients were in line with those defined by the National Transplant Center:-No previous HAART: CD4 T-cell counts ≥100 cells/µL and HIV viral load (VL) below the limit of detection.-Patient in HAART: CD4 T cell ≥100 cells/µL and HIV viral load (VL) below the limit of detection if no previous AIDS opportunist events or CD ≥200 if previous AIDS events.-No visceral Kaposi’s sarcoma or malignancies [13].

### 2.2. Selection Criteria

The preoperative diagnosis of HCC was based on the criteria defined by the European Association for the Study of the Liver (EASL) and the American Association for the Study of Liver Diseases (AASLD). The diagnosis of HCC was confirmed when the tumor characteristics were concordant with two imaging techniques, whereas biopsy of the tumor was proposed in the event of doubtful cases [14,15].

Patients were listed for transplantation if the tumor met the University of California-San Francisco Criteria: (1) a single tumor nodule with <6.5 cm diameter; (2) 2 or 3 nodules of ≤4.5 cm diameter and sum of the total diameter ≤8 cm; (3) no vessel or lymph node involvement or extrahepatic disease [16]. Over the last 5 years, in the process of evaluation for the listing of patients with HCC, the selection criteria were expanded by implementing the level of alpha-fetoprotein (AFP) as a biological marker of tumor aggressiveness [17].

In our region, the allocation policy has undergone several changes over the years in the strategy of allocating organs to transplantation centers. Currently, the system involves the allocation of organs to patients with a higher MELD score, although many exceptions to the MELD score are provided for specific diseases. Indeed, patients with HCC are assigned extra priority MELD points only in patients with Stage II (T2 one tumor <5 cm in diameter, 2 or 3 nodules <3 cm), while no points are awarded to patients with Stage I (T1 single lesion <2 cm in diameter) [18].

### 2.3. HCC Pre-LT Treatment

During the waiting list time, the multidisciplinary team had the task of proposing the most appropriate preoperative treatment for each HCC patient based on tumor staging and liver function. In patients with preserved liver function MELD < 9 or Child A/B, and absence of severe portal hypertension, minimally invasive hepatic resection was indicated in cases of single or multiple tumors but confined to a single segment or liver sector [19].

Patients with multifocal HCC or impaired liver function were treated with transarterial chemoembolization (TACE), whereas radiofrequency ablation (RFA) was offered in cases of single tumors less than 5 cm [20].

Patients with HCC that did not meet the standard criteria underwent a downstaging protocol with the intention of downsizing the tumor into the transplantability criteria. A three-month observation period was required to assess the response to downstaging treatment. Most of the patients listed for HCC, whose tumors were within the San Francisco Criteria, underwent local-regional treatment or hepatic resection in order to prevent tumor progression during the waiting period on the list (bridging treatment) [21,22].

All explanted livers were examined for total number of HCC nodules, diameter of the nodules, presence of microvascular invasion, lymph node invasion, satellites, and dysplastic nodules. Satellite nodules were considered as all the small foci of a tumor close to the main nodules, whereas dysplastic nodules were defined as precursors. The tumor degree of differentiation was defined according to Edmondson criteria [23].

Every patient on the waiting list was followed up clinically; this included an ultrasound scan (US) and measurement of serum AFP level every three months, and contrast-enhanced dynamic computed tomography (CT) or magnetic resonance imaging (MRI) every six months.

### 2.4. Surgical Procedure and Post-Operative Management

Liver transplantation was performed according to the piggyback technique, with different technical caval reconstructions, including end-to-end anastomosis on the three hepatic veins or end-to-side cavo-caval anastomosis. On occasion, a temporary portocaval shunt (TPCS) was used during the procedure to preserve the portal venous flow thereby reducing the intestinal congestion during the anhepatic phase [24]. The conventional technique without veno-venous bypass was only occasionally used.

Immunosuppression therapy (IS) was according to our standard protocol of dual-therapy with CNI and steroids, with or without a basiliximab induction. After receiving the CNI regimen for three to six months, many HCC LT patients were switched to treatment with an m-Tor inhibitor [25]. The antiretroviral therapy was discontinued during the peri-operative period to ensure that the concentrations of immunosuppressive drugs could be reliably predicted. HAART was reintroduced, on average, 2–5 days after LT, which is the time necessary for the stabilization of liver function parameters and blood IS concentrations [26].

### 2.5. Study Design and Statistical Analysis

The HCC HIV-infected LT group was compared with a control group consisting of the cohort of HCC liver transplant patients performed in the same period. The final study cohort therefore consisted of 386 HIV-uninfected LT and 34 HIV-infected LT.

Continuous variables in each group were compared with the independent sample T-test. Categorical parameters were compared using Fischer’s exact test. Survival rates after LT were calculated according to the Kaplan–Meier method and compared using the log rank test.

Recipient, operative, and donor variables of the entire study population were entered in a univariate analysis and multivariate analysis for patient survival using the Cox proportional hazard regression model. The variables used to build the Cox regression and univariate analysis are available to examine as Appendix A. All co-variates with *p* < 0.1 in the univariate analysis were included in a multivariate Cox regression analysis by forward conditional stepwise method. The logistic regression analysis was used to identify prognostic factors of HCC recurrence in the study population.

In order to overcome potential biases related to some differences of pre-operative variables between the two groups of the study, a propensity score matching analysis was also performed. The results of the analysis are available as a Appendix A.

Results are expressed as means with standard deviation. A *p* value of 0.05 or less was considered statistically significant. All the data analyses were carried out using SPSS 20 for Windows (release SPSS Chicago, IL, USA).

## 3. Results

A total of 32 HCC patients with HIV received 34 liver transplants. The baseline demographics and clinical characteristics of the HIV-infected population and the control uninfected group are summarized in Table 1.

Patients with HIV infection were younger, with a mean age at time of LT of 50.3 ± 5.6 years compared to patients in the HIV-uninfected group 57.6 ± 8.1, *p* = 0.001. The most frequent indication for LT was HCV and HBV-related cirrhosis. While the rate of HBV infection did not differ between the two groups (23.5% vs. 25.1%), the HCV infection was most frequently observed in the HIV+ group (70.5% vs. 54.1%). In the HIV-infected group, HCV-RNA was detectable at the time of transplantation in 38.2% of the HIV/HCV coinfected patients (n = 13).

The two groups in the study were homogeneous as regards the degree of severity of cirrhosis measured by the MELD score at the time of transplantation: 17.5 vs. 15.3 (*p* = ns) in the HIV+ vs. the HIV– groups. In addition, the Child -Pugh score was similar between the two groups; in particular, the rate of Child C stage was 35.2% vs. 33.1% in the HIV+ and HIV− groups.

The time on the waiting list was similar without statistically significant differences between the two groups: 150.3 ± 33 vs. 146.1 ± 9.6 days in the HIV-infected and HIV uninfected groups, respectively.

At the time of transplantation, HIV-infected patients had a mean CD4 T-cell count of 328 ± 214 cell/µL, while one patient had a detectable HIV viral load at the time of transplantation due to a temporary discontinuation of the HAART.

In the HIV+ group, one patient received a combined liver-kidney transplant while in the HIV− group, four underwent a combined liver-kidney transplant. One patient received a liver graft from an HIV positive donor. All patients were followed up for a mean 62 ± 51 months.

### 3.1. Graft and Patient Survival. Multivariate Analysis of Factors Impacting Survival Post-LT

Overall survival of the HCC HIV+ group vs. the HCC HIV− group at one, three, five, and 10 years was 84.3%, 80.6%, 71.6%, and 71.6% vs. 85.2%, 73.9%, 69.6%, and 61%, without statistically significant differences.

Graft survival at one, three, five, and 10 years was 81.1%, 77.4%, 68.3%, and 68.3 versus 83.7%, 71.9%, 68.2%, and 59.3% in the HIV+ and HIV– groups *p* = ns, respectively (Figure 1 and Figure 2).

Retransplantation (Re-LT) was necessary in two patients (5.8%) in the HIV+ group and in 18 patients (4.7%) in the HIV– group, *p* = ns. Post-transplantation 30-day hospital mortality was 2.9% (n= 1 patient) in the HIV+ group and 3.6% in the HIV– group (n = 14 patients), *p* = ns.

No statistically significant differences were observed in terms of peri and post-operative complications (Table 2).

A statistically significant improvement of overall survival was observed in the HIV-infected transplant group by dividing the transplant period into two historical eras: Era I 2003–2011 (n = 17 patients) vs. Era II 2012–2020 (n = 16 patients). The 5-year survival rate was 47.1% and 92% respectively.

At multivariate Cox regression analysis, the independent predictive factors of survival in the entire population HIV+ and HIV– groups, were: HCV infection HR 1.83 *p* = 0.024 (Table 3).

### 3.2. Pathological Characteristic of Tumors of the Study Populations

Table 4 shows the pathological tumor characteristics of the study population. In the HIV-infected HCC group, preoperative treatments were used in 26 patients (76.4%): 14 patients had trans-arterial chemoembolization (TACE), two had radiofrequency ablation (RFA), five patients had combined trans-arterial chemoembolization and radiofrequency, and two had combined trans-arterial chemoembolization and alcohol injection (PEI); three patients had previously undergone hepatic resection. One of the latter patients received the liver transplant as a salvage transplantation for tumor recurrence, whereas the other two patients were transplanted for “high risk” HCC as “de principe salvage liver transplantation”. In the HIV-uninfected group, 310 patients (80.4%) had preoperative treatment of HCC during the waiting list time: 148 patients had trans-arterial chemoembolization (TACE), 66 underwent radiofrequency ablation, 42 had combined TACE and radiofrequency ablation, and 54 had hepatic resection. No difference was observed between the two groups in terms of rate of HCC treatment pre-LT.

The mean number of HCC nodules was slightly higher in the HIV+ group (2.7 ± 2.4) than in the HIV− group (2.3 ± 1.8), *p* = 0.024, while no statistically significant difference was observed in the two groups regarding the mean diameter of the largest HCC nodule, which was 2.4 ± 1.7 cm vs. 3 ± 2.1 cm in the HIV+ and HIV– groups respectively.

In 16 patients (47%) in the HIV+ group the tumor was unifocal, while 158 patients (40.9%) in the HIV- group had a unifocal HCC.

There were no significant differences between the two groups regarding grading (G1/G2 73.5% vs. 82.1% in HIV+ vs HIV- groups) nor rate of microvascular invasion (14.7% vs. 16.8% in HIV+ vs. HIV– groups). At post-transplant pathology 11 patients (32.4%) in the HIV-infected group and 110 patients (28.5%) in the control group were outside the UCSF criteria.

### 3.3. Multivariate Analysis of Predictors Factors Impacting HCC Recurrence Post-LT

HCC recurrence developed in five patients (14.7%) in the HIV+ group and in 49 patients (12.6%) in the HIV- group during the follow-up. The median time from LT to HCC recurrence was 27.6 months (range 6.1–117.5 months). In the HIV-infected group, tumor recurrence in most cases was principally intrahepatic in three patients, while two patients developed recurrence in the adrenal glands.

The logistic regression analysis of the study population demonstrated that microvascular invasion HR 5.18 *p* < 0.001, HCC diameter (the largest nodule) HR 1.16 *p* = 0.028, and number of HCC nodules HR 1.26 *p* = 0.003 are predictors of recurrence post-LT (Table 3).

The logistic regression analysis in the HIV+ group after transplant showed that a high grade of differentiation (Grading 3–4) was an independent predictor of recurrence, HR 15.2 *p* = 0.026.

Disease-free survival at one, three, five, and 10 years was 93.6%, 88.6%, 86.9%, and 85% vs. 92.3%, 88.1%, 88.1%, and 79.3% in the HIV− and HIV+ transplant groups respectively, *p* = 0.46.

## 4. Discussion

In the pre-HAART Era, HIV infection was considered a contraindication for liver transplantation, due to worse outcomes in infected patients than those observed in the uninfected population. Furthermore, the concern that immunosuppressive therapy could worsen the risk of post-transplantation opportunistic infections, and therefore the risk of loss of the graft, had led most transplant centers to severely limit access to LT in HIV+ candidates [27,28].

The introduction of HAART, the improvement of post-LT clinical management, and more precise use of immunosuppressive therapy have made it possible to achieve incredible results: liver transplantation in HIV-infected patients now appears to be a well-established indication in most transplant centers [29,30,31].

However, if we carefully analyze many of the studies published in the literature on this topic, the survival of HIV-infected patients appears to be lower than in the uninfected group. Data from an American registry in which 180 HIV positive LT patients were compared with the HIV-matched control group show that the overall survival at five years in HIV+ patients was statistically lower than in the HIV negative group (55.8% versus 73.4%) [32].

In our study involving one of the largest cohorts of HIV-infected patients undergoing liver transplantation for HCC and followed with an exceptionally long follow-up, we demonstrate instead that the overall and graft survival of the two groups is not statistically different.

In a recent study based on a European and American registry on 658 HIV-infected LT patients, the HIV infection was confirmed at multivariate analysis as an independent predictive factor of graft loss (HR 1.41 *p* = 0.001) and patient death (HR 1.61 *p* = 0.001) [33].

Unlike this study, we showed instead that HIV infection is not a negative prognostic factor of post-LT survival at multivariate analysis.

This finding is also particularly evident if we compare the first era with the second part of the study in which the survival of HIV+ LT recipients appears a further improved 5-year survival rate (47% vs 92%).

There are several factors that may contribute to the improvement in survival observed over the past few years.

The new retroviral drugs have certainly made the post-LT pharmacological therapy easier to manage, owing to the reduced interactions of these drugs with immunosuppressants [34]. Protease inhibitors and nonnucleoside reverse transcriptase inhibitors had a severe effect on calcineurin inhibitors and mTor inhibitor levels that did not allow immunosuppressive therapy to reach adequate pharmacological levels. Such imbalances were responsible for the high rate of acute cellular rejection (ACR) in earlier series [35]. However, our study did not show different rates of ACR between the two groups in the study over time. New HIV treatment regimens that include integral inhibitors have become the recommended HART regimen in the transplant setting as they do not interact with standard maintenance immunosuppressants even when HCV treatment is co-administered [26].

Although the best immunosuppressive therapy in HIV-infected patients has not yet been proven, sirolimus and everolimus (mTor drugs) have emerged as very promising immunosuppressant agents in the management of post-LT therapy. As indicated by our previous work, these drugs have an intrinsic anti-proliferative and antiviral activity, have a reduced interference with HAART and at the same time keep the risk of nephrotoxicity and neutoxicity low, so they appear to be particularly useful as immunosuppressant drugs in this setting of patients [25]. However, mTor inhibitors have not been shown to have a direct impact on post-LT survival at multivariate analysis of our study.

Patient and graft survival post-LT were significantly lower in HCV/HIV-coinfected patients than in HCV monoinfected recipients, due to the rapid progression of fibrosis and a higher incidence of severe forms of recurrence of HCV infection [36,37]. Our study also shows how HCV infection at the time of transplantation represents a negative prognostic factor of post-LT survival.

The advent of direct acting antivirals (DAAs) has resulted in significant improvements in the treatment of HCV recurrence in patients with coinfection. Several large trials in coinfected individuals have demonstrated sustained virologic response (SVR) rates >93% with a variety of DAA agents. The same result has recently been replicated in a coinfected HIV/HCV population undergoing liver transplantation where an SVR rate of 94% was obtained [38,39,40]. However, the impact of DAA in the prognosis of HCV positive patients at the time of transplantation and treated with DAA in post LT has not yet been adequately studied. In our study, five patients were treated with DAA post LT but SVR did not show any impact on the survival or recurrence of HCC, but this should be interpreted bearing in mind the small number of patients treated.

HIV infection complicated by HCV or HBV coinfection seems to greatly precipitate in the development of HCC due to accelerated carcinogenicity, although the real pathogenic mechanism has not been fully understood [41,42,43,44,45]. Currently, it has been clearly shown that 25% of liver death in HIV+ patients is linked to the presence of HCC [46,47]. In the study by Puoti et al. it was explained that HIV-infected patients with HCC can present with a more advanced stage, with a higher rate of infiltrating forms and, sometimes, with extrahepatic-extranodal metastases [48].

HIV-infected patients with HCC have a dismal survival rate with respect to HIV-negative patients. However, in a retrospective single-center study comparing 63 positive HIV+ patients with HCC and 226 HIV– patients in a period between 1992 and 2005, it was shown that 5-year survival in both groups was not significantly different (17% versus 22%) [49].

Therefore, while liver transplantation in HIV-infected subjects is considered a clinical indication consolidated by the encouraging results published over the years, the impact of HCC on the post-LT outcome has been little explored in the HIV-infected population [50].

A recent Italian multicenter study analyzed 30 HIV-infected patients with HCC comparing them with a matched controlled group of 125 HIV-uninfected subject. The authors reported a three-year survival of 65% versus 60%, *p* = 0.33, and a recurrence rate of 6.7% versus 14.8%, *p* = 0.15 [51].

Our study confirms that HCC HIV+ patients can achieve long overall survival and disease-free survival at five years post-LT similar to that of uninfected patients.

Although it is known that HIV-infected patients have a higher risk of dropout while on the waiting list, none of our HIV patients received further extra prioritization during the time on the waiting list, so there was no difference between the two groups in the study in time spent on the waiting list. A French study recently highlighted that, based on intention-to-treat analysis, survival after LT for HCC was impaired by HIV infection. In this study, survival rates after listing at three years were 55% and 82% in HIV+ and HIV- patients, respectively (*p* = 0.0005) [52].

The majority of our HIV+ patients were started on systematic treatment of HCC prior to the transplantation both with locoregional therapy (LRT) or minimally invasive surgical resection of HCC in order to avoid tumor progression while on the waiting list. Although our patients had undergone bridging or down-staging treatment of HCC, at post-transplant pathology 32.4% of the patients had an HCC outside the listing criteria (UCSF criteria) and 20% of patients were outside the Up-to-seven criteria (extended Milan criteria), probably due to an underestimation of the HCC tumor at pre-LT imaging. The subgroup analysis of HIV+ patients with HCC outside the UCSF criteria achieved a survival rate at five years of 24% [53]. Therefore, according to our results, thinking of extending the selection criteria for LT in these patients is very risky since the probability of recurrence seems excessively high. Nevertheless, as demonstrated by our study, using only the static criterion of the number and size of the tumors appears to be completely ineffective in stratifying the risk of post-LT recurrence.

In fact, in our study the only prognostic factor in the group of HIV+ patients capable of identifying a high risk of recurrence post-LT was a high-grade HCC (G3-G4). Although this is based on post-LT pathological data, some groups have systematically proposed biopsy of the nodule before placing patients on the waiting list and denied LT to patients with G3 cancer [54,55].

## 5. Conclusions

In conclusion, our study, the largest single-center study ever performed in this field, showed that HIV patients with HCC now have post-LT results comparable to uninfected patients. Careful management of HCC from its onset and throughout the time on the waiting list, involving the use of loco-regional or surgical treatments, is the key to success for HIV-infected patients with HCC.

## Figures and Tables

**Figure 1 cancers-13-04727-f001:**
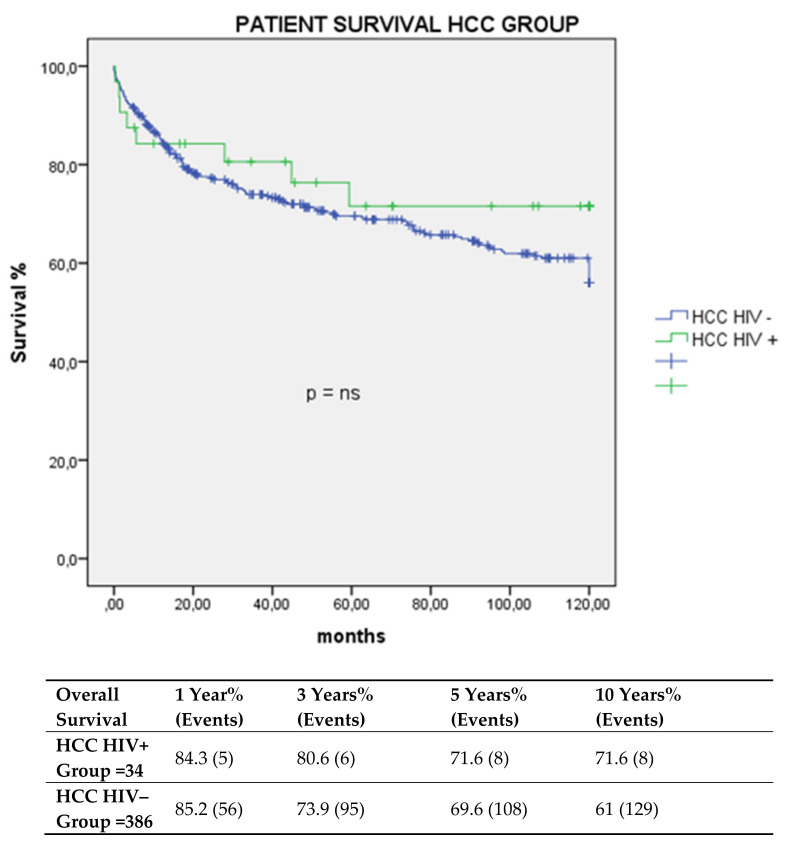
Overall survival in HCC trasplant group.

**Figure 2 cancers-13-04727-f002:**
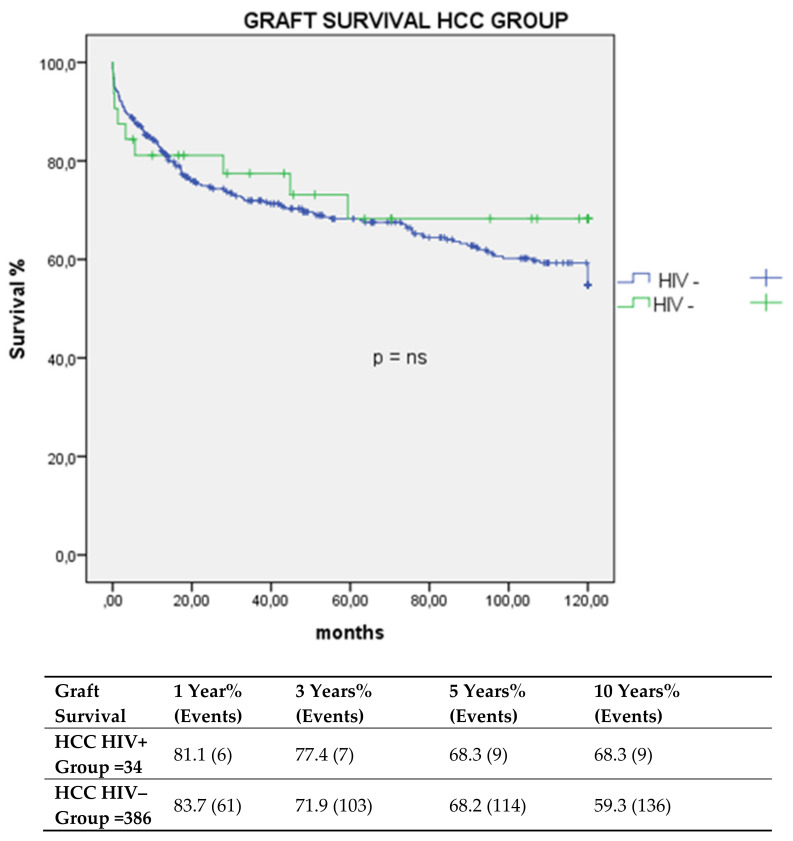
Graft survival in HCC transplant group.

**Table 1 cancers-13-04727-t001:** Demographic characteristics of the study population.

Study Population = 420						
Variable		HIV Group *n* = 34		Control Group *n* = 386		*p* Value
		Number	%	Number	%	
Sex	M	30	88.2	320	82.9	ns
	F	4	11.8	66	17.1	
Age	mean and sd	50.3	±5.6	57.6	8.1	0.001
Child	A/B	22	64.8	258	66.9	ns
	C	12	35.2	128	33.1	
Meld at LT	mean and sd	17.5	±9.6	15.3	±7.2	ns
CD4	mean and sd	328	±214			
BMI	mean and sd	23.9	±3.7	26	6	ns
Type of LT	Isolated	33	97	382	98	ns
	Combined Kidney-Liver	1	3	4	2	
Etiology	HCV	24	70.5	209	54.1	0.019
	HBV	8	23.5	97	25.1	ns
	Alcohol	2	6	44	11.4	ns
	Biliary	0	3	3	0.8	ns
	Metabolic	0		22	5.7	ns
	Other	0		11	2.9	ns
Waiting list time days	mean and sd	150.3	±33	146.1	±9.6	ns
Re-OLT	yes	2	5.8	18	4.7	ns

**Table 2 cancers-13-04727-t002:** Post Liver transplant complications in the study population.

Post-Transplant Complications Study Population = 420						
Variable		HIV Group *n* = 34		Control Group *n* = 386		*p* Value
		Number	%	Number	%	
Acute Cellular Rejection	yes	6	17.6	45	11.6%	ns
Biliary Stenosis	yes	7	20.5	42	10.8	ns
Early Hepatic Artery Thrombosis	yes	1	2.9	10	2.5	ns
Late Hepatic Artery Thrombosis	yes	1	2.9	12	3.1	ns
Portal Vein Thrombosis	yes	0	0	9	2.3	ns
De Novo Tumor	yes	1	2.9	26	6.7	ns

**Table 3 cancers-13-04727-t003:** Multivariate analysis of predictor factors of survival and recurrence post-Lt.

Multivariate Analysis					
Variable	B	S.E.	HR	95% C.I. for EXP	*p* Value
Cox regression of overall survival					
HCV etiology	0.608	0.268	1.837	1.085–3.109	0.024
HIV Infection	0.72	0.51	2.06	0.75–5.65	ns (0.159)
Logistic regression for HCC recurrence in the study population					
Numbers of HCC nodules	0.233	0.079	1.262	1.081–1.474	0.003
Diameter of the largest nodule	0.152	0.069	1.164	1.017–1.333	0.028
Microvascular invasion	−1.645	0.455	5.18	2.12–12.64	0.001
HIV infection	−0.107	0.77	0.89	0.195–9.130	ns (0.88)

**Table 4 cancers-13-04727-t004:** Tumor characteristics of the study population.

HCC LT POPULATION =420						
Variable		HIV Group Number = 34		Control Group Number = 386		*p* Value
HCC Diameter	mean and sd	2.4	±1.7	3	±2.1	ns
HCC N. Nodules	mean and sd	2.7	±2.4	2.3	±1.8	0.024
α-FETO-Protein ng/mL	mean and sd	65	±14.1	68	±186	ns
HCC Focality	uni	16	47%	158	40.9%	ns
	multi	18	53%	228	59.1%	
HCC Treatment Pre-LT Downstaging	no	8	23.6%	76	19.6%	ns
	yes	26	76.4%	310	80.4%	
Microvascular Invasion	no	29	85.3%	321	83.2%	ns
	yes	5	14.7%	65	16.8%	
Capsule Invasion	no	28	82.3%	340	88 %	ns
	yes	6	17.7%	46	12%	
Satellitosis	no	28	82.3%	353	91.4%	ns
	yes	6	17.7%	33	8.6%	
Grading	G1–2	25	73.5%	317	82.1%	ns
	G3–4	9	26.5%	69	19.9%	
Milan Criteria	In	21	61.7%	245	63.4%	ns
	out	13	38.2%	141	36.6%	
USCF	In	23	67.6%	276	71.5%	ns
	out	11	32.4%	110	28.5%	
Up-to Seven Criteria	In	27	79.4%	297	76.9%	ns
	out	7	20.6%	89	23.1%	
Recurrence	yes	5	14.7%	49	12.6%	ns
	no	29	85.3%	337	87.4%	

## Data Availability

Data are contained within the article.

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
