# Peer review of "Liver Transplantation for HCC in HIV-Infected Patients: Long-Term Single-Center Experience"

_cancers, 2021, doi:10.3390/cancers13184727_

Round 1
Reviewer 1 Report
This paper has improved substantially with the modifications introduced in the different sections.
Author Response
To Reviewer 1
I thank you for your positive comment and I am glad you liked the article.
Best regards
Gian Piero Guerrini MD PhD FACS FEBS
Reviewer 2 Report
Authors re-analyzed their study subject also by using propensity score matching according to the reviewer 2. And its results revealed similar results and conclusions as their previous analysis.
Table 3 must be the result of multivariate analysis for predicting survival and recurrence after liver transplantation in all patients who underwent liver transplant of HIV+ and HIV – groups.
There were no significant difference as a predictor in HIV factor. It should be shown in Table 3 that HIV positivity did not reveal any significant difference as NS for clear understanding.
Author Response
To reviewer 2,
I thank you for your positive comment. I have modified Table 3. As you suggested I showed how HIV infection is not an independent prognostic factor of survival and recurrence
Best regards
Gian Piero Guerrini Md PhD FACS FEBS
Reviewer 3 Report
HCC is becoming a leading cause of morbidity in HIV-infected patients since highly active antiretroviral therapy induces long-life expectations by reducing AIDS-related complications. In the present investigation, Guerrini et al. assessed the results of liver transplantation (LT) for HCC in HIV-infected patients. The authors found that HIV patients undergoing LT for HCC display comparable data in terms of post-LT survival to uninfected patients. Therefore, HIV-infected patients with HCC can achieve prolonged survival as long as close surveillance and precise tumor treatment are employed before LT.
The study by Guerrini et al. is novel, well written, and provides relevant insights on the management and prognosis of HIV-infected patients undergoing LT for HCC. This is a critical and increasing issue in the clinics. The study was properly conducted, and the data were correctly analyzed, supporting the conclusions drawn. Figures and tables are clear and easy to understand.
I have no concerns on this study.
Author Response
To Reviewer 3
I thank you for your positive comments and I am glad you liked our article.
Best regards
Gian Piero Guerrini MD PhD FACS FEBS
This manuscript is a resubmission of an earlier submission. The following is a list of the peer review reports and author responses from that submission.
Round 1
Reviewer 1 Report
This review paper is of great interest in the field of liver transplantation as it confirms that the presence of an HIV infection does not influence the survival of the transplanted patient. However, in my opinion, the results are extensive and can be summarized by some table with graft and patient survival. Finally, in transplants patients with hepatocellular carcinoma , has Sirolimus been used as an immunosuppressive therapy?.
Author Response
RESPONSE TO REVIEWER 1 COMMENTS
This review paper is of great interest in the field of liver transplantation as it confirms that the presence of an HIV infection does not influence the survival of the transplanted patient. However, in my opinion, the results are extensive and can be summarized by some table with graft and patient survival. Finally, in transplants patients with hepatocellular carcinoma , has Sirolimus been used as an immunosuppressive therapy?.
We are glad the article was of interest to you. We thank you for your precise and careful remarks that we have taken into account in modifying our article in order to make it useful to the scientific community and readers of the Cancer journal
Just as you pointed out, the message we want to bring out from this study is that patients with HIV suffering from ESLD and/or HCC have a huge benefit in accessing the transplant program. The survival of these patients is now comparable to the group of non-infected patients as long as they are managed correctly from the immunological and infectious point of view in the peri- post transplant period. Particular attention should be paid to HCC patients who need to be treated with aggressive downstaging and bridging programs during the waiting time on the list.
We have used Sirolimus many times in patients with HIV+ but also not HIV non-infected. We consider the drug very useful for its known antiproliferative, antiviral properties, keeping the risk of nephrotoxicity low. In our cohort of LT HIV+ patients, 15 are on Sirolimus as the only immunosuppressive drug. We have added this to the discussion section.
Best Regards
Gian Piero Guerrini MD Phd

Reviewer 2 Report
The authors describe a large cohort of patients after liver transplantation from a single center with the focus on HIV infected patients and HCC.
The cohort is interesting, however, my main and severe concern is that it is unclear what the goal of the authors is: The paper jumps between HIV aspects and HCC aspects. It is unclear why you focus on the HCC aspect in the HIV+ and HIV- groups.
The differences in age and etiology of the HIV group compared to the rest of the cohort are a concern for further analysis. I would suggest propensity score matching to overcome the problem.
There are several inconsistencies in the paper: 1) in the abstract 65 patients with HIV are mentioned, in the results these are 63 patients with two retransplantations leading to 65 grafts. 2) The inclusion criterium for HCC in this center are the UCSF criteria, in the text you the report numbers of in/outside MIan criteria (in the results) and UCSF and up-to-seven in the discussion. 3) In Table 4 it is unclear if radiological or histological criteria are shown.
It is unclear from the text what the criteria were to distinguish between era 1 and era 2 in HIV. The sample size in these two groups is very small, so conclusions need to be reported with caution.
Minor points:
How many HIV+ patients received an HIV+ graft?
Please report how many patients with HCV were cured with DAA after transplantation - does this impact on the HCC recurrence rate?
What is the merit of table 1? This can be shown in the text as well.
HCC is not an etiology of liver disease (table 4) it should be reported separately from etiology.
What do you mean with "systematic treatment of HCC" -please explain
What do you mean with "the selection criteria were expanded by implementing the lev‐ 95 el of alpha‐fetoprotein (AFP) as a biological marker of aggressiveness" - please explain
"discarding patients" is a very disrespectful term, please rephrase
Please carefully edit your paper for spelling (several words are capitalized; "positive HIV+" is an unnecessary iteration, Milan or Milano criteria? ...)
Author Response
Response to Reviewer 2 Comments
The authors describe a large cohort of patients after liver transplantation from a single center with the focus on HIV infected patients and HCC.
The cohort is interesting, however, my main and severe concern is that it is unclear what the goal of the authors is: The paper jumps between HIV aspects and HCC aspects. It is unclear why you focus on the HCC aspect in the HIV+ and HIV- groups.
The differences in age and etiology of the HIV group compared to the rest of the cohort are a concern for further analysis. I would suggest propensity score matching to overcome the problem.
There are several inconsistencies in the paper: 1) in the abstract 65 patients with HIV are mentioned, in the results these are 63 patients with two retransplantations leading to 65 grafts. 2) The inclusion criterium for HCC in this center are the UCSF criteria, in the text you the report numbers of in/outside MIan criteria (in the results) and UCSF and up-to-seven in the discussion. 3) In Table 4 it is unclear if radiological or histological criteria are shown.
It is unclear from the text what the criteria were to distinguish between era 1 and era 2 in HIV. The sample size in these two groups is very small, so conclusions need to be reported with caution.
Dear Reviewer
We thank you for your precise and careful remarks that we have taken into account in modifying our article in order to make it useful to the scientific community and readers of the journal Cancer.
We tried to write an article that would make readers understand that patients with HIV and ESLD and/or HCC have a huge benefit in accessing the transplant program. However, most of the transplant centers that offer liver transplantation to HIV patients have lower survival results than uninfected HIV patients. That is why we wrote an article that made the readers understand the multiple aspects of this complex pathology. Our study deals with the transplantation of both HIV in general and HIV patients with HCC.
We have tried to focus the study by bringing out the strength of this cohort of 63 patients transplanted into HIV patients, this population was very homogenous and was followed with a very long follow-up.
We recalculated the pathology of patients with HCC, as you suggested, not considering the tumor as etiology (reviewer: "HCC is not an etiology of liver disease (table 4) it should be reported separately from etiology"). This made it possible to demonstrate instead that HCV positivity between the two groups does not constitute a significant difference, thus reducing the possibility of bias. This change was updated in Table 1.
We have corrected all the inconsistencies that you detected in the paper:
- a) "in the abstract 65 patients with HIV are mentioned, in the results these are 63 patients with two retransplantations leading to 65 Grafts"
We have specified in the abstract that these are 63 patients who received 65 liver grafts and this has been further explained in the methods.
b)"The inclusion criterium for HCC in this center are the UCSF criteria, in the text you the report numbers of in/outside Mian criteria (in the results) and UCSF and up-to-seven in the discussion".
We modified the sentence in the results section by reporting the data outside UCSF; in the discussion we used the term extended Milan Criteria.
- c) "In Table 4 it is unclear if radiological or histological criteria are shown".
We have modified the text by writing: Table 4 shows the pathologic tumor characteristics of the HCC LT study population.
Minor points:
- How many HIV+ patients received an HIV+ graft?
Two patients received a positive HIV graft; this was explained in the results section
- Please report how many patients with HCV were cured with DAA after transplantation - does this impact on the HCC recurrence rate?
12 patients were treated with DAA; survival or recurrence at univariate analysis Fischer exact test did not differ in this group of patients compared to untreated patients
- What is the merit of table 1? This can be shown in the text as well.
We have deleted Table 1 and written it in the text of the article
5)What do you mean with "systematic treatment of HCC" -please explain
This term refers to an organized, repetitive and reproducible procedure aimed at treating the tumor. It refers to a very precise system of treatment, therefore methodical. If these words could be misinterpreted, we are ready to delete them
6)What do you mean with "the selection criteria were expanded by implementing the level of alpha‐fetoprotein (AFP) as a biological marker of aggressiveness" - please explain
Many authors have developed new transplant selection criteria for HCC patients, integrating tumor staging (size and number) and alpha-fetoprotein value. We also apply these criteria increasingly in our patients with HCC who are candidates for LT. This strategy is mainly used to bring patients back within the UCSF criteria through a suitable dowstanging program. We have included the appropriate reference in the text.[1, 2]
7)”discarding patients" is a very disrespectful term, please rephrase
We probably expressed ourselves badly and changed the word
8) Please carefully edit your paper for spelling (several words are capitalized; "positive HIV+" is an unnecessary iteration, Milan or Milano criteria? ...)
We understand your observation and reduced the capitalized letters by also postponing the final editing layout to the journal. Milan Criteria is the correct term.
Best Regards
Gian Piero Guerrini Md PhD
- Duvoux C, Roudot-Thoraval F, Decaens T, Pessione F, Badran H, Piardi T, Francoz C, Compagnon P, Vanlemmens C, Dumortier J et al: Liver transplantation for hepatocellular carcinoma: a model including α-fetoprotein improves the performance of Milan criteria. Gastroenterology 2012, 143(4):986-994 e983; quiz e914-985.
- Mazzaferro V, Sposito C, Zhou J, Pinna AD, De Carlis L, Fan J, Cescon M, Di Sandro S, Yi-Feng H, Lauterio A et al: Metroticket 2.0 Model for Analysis of Competing Risks of Death After Liver Transplantation for Hepatocellular Carcinoma. Gastroenterology 2018, 154(1):128-139.

Round 2
Reviewer 2 Report
The authors answered all my minor comments sufficiently but did not respond to my major concerns:
1) The inadequate matching of the study populations
2) The authors explained in their letter why they mixed the two topics of HIV and HCC in liver transplantation but did not make any attempts to restructure their manuscript. For an oncological journal such as "Cancers" I would suggest to completely restructure the paper to put HCC in the focus and add HIV as an additional aspect.